# Clinical feature and gene expression analysis in low prostate-specific antigen, high-grade prostate cancer

**Peng Zhang** *, **Tieding Chen, Ming Yang**

Department of Urology, Ningbo Medical Center Lihuili Hospital, Ningbo, Zhejiang, China

* loversdrug@gmail.com

**Data availability statement:** The data used in this study come from two sources. The first

## Abstract

### Background

Prostate cancer (PCa) patients with low prostate-specific antigen (PSA) levels can occasionally present high-grade disease. These patients often exhibit resistance to androgen deprivation therapy and have poor outcomes. The mechanisms underlying these observations remain poorly understood. This study aimed to investigate the clinical characteristics and potential gene expression mechanisms in this subgroup.

### Patients and Methods

Clinical data from 365,558 PCa patients were categorized into four groups based on PSA levels and Gleason score (GS): Group 1 (PSA ≤ 2.5 ng/mL, GS < 8), Group 2 (PSA ≤ 2.5 ng/mL, GS ≥ 8), Group 3 (PSA > 2.5 ng/mL, GS < 8), and Group 4 (PSA > 2.5 ng/mL, GS ≥ 8). Clinical characteristics were compared using Kruskal-Wallis H and Pearson's chi-squared tests. Competing-risks regression assessed prostate cancer-specific mortality (PCSM). Gene set enrichment analysis (GSEA) was performed on 219 PCa patients to compare Group A (PSA ≤ 2.5 ng/mL, GS ≥ 8) with Group B (PSA > 2.5 ng/mL, GS ≥ 8).

### Results

Group 2 had a significantly higher tumor stage (p < 0.001) and increased hazard ratio for PCSM (p < 0.001). GSEA in Group A identified 156 upregulated gene sets and highlighted several enriched pathways, including the polycomb repressive complex 2, the epidermal growth factor receptor family, retrograde axonal transport, the tumor necrosis factor/nuclear factor-κB pathway, the Rho guanine nucleotide exchange factor/RhoA pathway, and the phosphoinositide 3-kinase signaling pathways (p < 0.05, false discovery rate-adjusted p < 0.25).

### Conclusion

PCa patients with low PSA levels and high GS demonstrated an increased risk of PCSM. They were characterized by the aberrant activation of multiple signaling pathways.

dataset is the GSE62116 dataset, which can be accessed through the Gene Expression Omnibus database at the following URL: https://www.ncbi.nlm.nih.gov/geo/geo2r/?ac-c=GSE62116. The second dataset is derived from the SEER 17 registry (2000-2019). Access to the SEER database is restricted to registered researchers.

**Funding:** The author(s) received no specific funding for this work.

**Competing interests:** The authors have declared that no competing interests exist.

Targeted therapeutic strategies aimed at these pathways warrant further investigation for their potential to improve outcomes in this aggressive PCa subtype.

## Introduction

Prostate cancer (PCa) is the most common malignancy in American men, accounting for 299,010 new cases and 35,250 deaths in 2024[1]. Its risk stratification scheme typically includes age, race, serum prostate-specific antigen (PSA) level, TNM stage, and Gleason score (GS)[2]. The primary treatment of nonmetastatic PCa includes radical prostatectomy (RP) and a combination of radiation therapy (RT) and androgen deprivation therapy (ADT)[2].

The development of PCa is known to be highly dependent on androgen[3]. After entering the cytoplasm, androgen binds to the ligand-binding domain of the androgen receptor (AR) and translocates to the nucleus. Through its DNA-binding domain, the AR recognizes and binds to androgen response elements in DNA. With the help of coactivators, it facilitates the activation of target genes, including PSA and transmembrane protease serine 2[4]. As a result, PSA has been recommended by multiple clinical guidelines as a routine marker for early detection and recurrence of PCa [2,5].

However, there are still instances in which patients with low PSA levels present high GS and have poor prognosis. Brandon A. Mahal *et al.* selected 494,793 patients with cT1-4N0M0 PCa, excluding those with neuroendocrine or small-cell histology. They found that 3,409 patients (0.69%) exhibited PSA ≤ 2.5 ng/mL and GS ≥ 8. The all-cause mortality (ACM) in this subgroup was significantly higher than that in the subgroup with PSA 4.1–10 ng/mL and GS ≥ 8[6]. Christian D. Fankhauser *et al.* conducted a study that included 102,089 patients diagnosed with cT1-T4N0M0 PCa. Within this cohort, 1,219 patients (1.19%) exhibited PSA ≤ 5 ng/mL and GS ≥ 8. In the high-grade subgroup, four-year prostate cancer-specific mortality (PCSM) was higher in patients with PSA ≤ 5 ng/mL than in those with PSA 5.1–10 ng/mL. This increased risk of mortality was observed in men treated with external beam RT, and in those who did not receive radical local treatment[7].

The underlying causes of this phenomenon remain unclear. Although previous studies have suggested that these tumors frequently exhibit higher expression of neuroendocrine/small-cell markers, both chromogranin A (CgA) staining in pathological sections and levels of CgA and neuron-specific enolase in serum samples appeared to be unrelated to PSA levels[6,8,9].

Consequently, the aim of this study was to clarify the clinical characteristics and prognostic implications of low-PSA, high-grade PCa in comparison to other subgroups, and to investigate its unique gene expression features and underlying causes.

## Patients and methods

### SEER cohort

The Surveillance, Epidemiology, and End Results (SEER) program provides information on cancer incidence and prognosis for 48% of the US population[10]. For this retrospective study, we accessed the SEER database on April 24, 2024. This study utilized the SEER 17 registry database (2000–2019) with inclusion criteria specifying "Site and Morphology" as the Primary Site-labeled = "C61.9-Prostate gland" and Site recode ICD-O-3/WHO 2008 = "Prostate". A total of 1,063,471 individuals were initially included. Clinical data collected included age, race, PSA level, histology, primary and secondary Gleason pattern, TNM classification, surgery status, radiotherapy status, surgery/radiotherapy sequence, chemotherapy status, cause of death,

and survival month. The exclusion criteria were incomplete data, T0 staging, and histology of neuroendocrine (ICD-O-3 8244/3, 8246/2, 8246/3), neuroendocrine differentiation (ICD-O-3 8574/2, 8574/3), small cell (ICD-O-3 8041/3, 8045/3), or large cell (ICD-O-3 8014/3) carcinoma. Ultimately, 365,558 individuals were included in this cohort. The screening process is illustrated in Fig 1.

It is important to note that PSA values recorded in the SEER database have been rigorously audited for accuracy since 2004[11]. Follow-up period was recorded as the period from the last day of available survival information or the day of diagnosis to death.

## Statistical analysis

Based on PSA levels and GS, the demographic and clinical data were categorized into four groups: Group 1 (PSA ≤ 2.5ng/mL, GS < 8), Group 2 (PSA ≤ 2.5ng/mL, GS ≥ 8), Group 3 (PSA > 2.5ng/mL, GS < 8), and Group 4 (PSA > 2.5ng/mL, GS ≥ 8). The Kruskal-Wallis test was used to compare continuous and ordinal data, while the Pearson's chi-squared test was used to compare categorical data.

Fine-Gray competing-risks regression analysis was utilized to calculate the hazard ratio (HR) in both univariate and multivariate analyses for PCSM. PCSM was defined as any death in which PCa was identified on the death certificate as a part of the sequence leading to death. Death from a cause other than PCa was considered a competing event when analyzing PCSM. The variables included in the analyses were age (continuous), race (non-black [referent], black), PSA and GS group (Group 1, Group 2 [referent], Group 3, Group 4), T stage (T1 [referent],

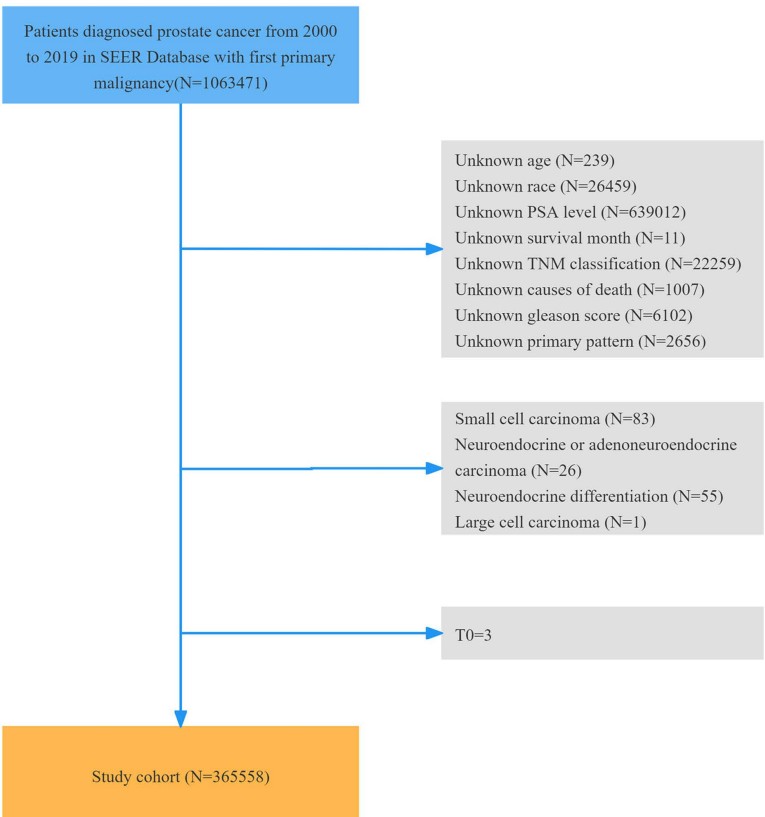

**Fig 1. Flow chart of the patient cohort selection.**

T2, T3, T4), N stage (N0 [referent], N1), M stage (M0 [referent], M1), surgery (no [referent], yes), RT (no [referent], yes), and chemotherapy (no [referent], yes). The Wald chi-squared test was used to assess the goodness of fit of the model. Cumulative incidence functions of the four groups for PCSM were subsequently generated and compared using Pepe and Mori tests.

Statistical analyses were performed using two-sided tests. The significance threshold was set at $p = 0.05$. For multiple comparisons, the significance level was adjusted to $p = 0.0083$ following the Bonferroni correction. Analyses were performed using Stata/MP 17.0 (StataCorp, College Station, Texas, USA).

## GEO cohort

To compare gene expression levels between two groups of PCa cases, namely those with PSA ≤ 2.5 ng/mL and GS ≥ 8 (Group A), and those with PSA > 2.5 ng/mL and GS ≥ 8 (Group B), we accessed the Gene Expression Omnibus (GEO) database (https://www.ncbi.nlm.nih.gov/geo/) on May 16, 2024. The GSE62116 dataset was retrieved to perform this retrospective analysis. This dataset encompasses RNA profiles from formalin-fixed, paraffin-embedded primary PCa specimens obtained from patients who underwent RP. It included a total of 219 specimens, each pathologically confirmed as prostate adenocarcinoma. Patients with metastasis at diagnosis or any prior treatment for PCa were excluded from the analysis.

## Gene set enrichment analysis

$Log_2$ fold change (FC) values were calculated for each gene between Groups A and B, and gene set enrichment analysis (GSEA) was conducted based on the $log_2$FC values using the Bioinformatics platform (https://www.bioinformatics.com.cn/), with reference to the KEGG MEDICUS subset of the CP gene set from the MSigDB human database. To minimize redundancy in the results, we utilized the aPEAR package to analyze similarities between gene sets and visualized the results as networks of enriched pathways, each labeled with a descriptive name to represent its biological significance[12]. We applied a significance level of $p < 0.05$, and a false discovery rate adjusted $p < 0.25$ for statistical analysis. Probe IDs were mapped to official gene symbols using the DAVID Bioinformatics Resources (https://david.ncifcrf.gov/summary.jsp). Probe IDs that did not correspond to any gene name were excluded from the analysis.

## Ethics statement

This study was conducted using the SEER 17 registry database (2000–2019) and the GSE62116 dataset, both of which are publicly available and do not require patient informed consent. Consequently, the local Ethics Committee approved an exemption from ethical review (Approval Number: KY2023ML031). During and after data collection, we did not have access to information that could identify individual participants.

## Results

### Clinical characteristics comparison

The demographic and clinical characteristics of the SEER cohort are presented in Table 1. The median follow-up period was 52 months. The clinical characteristics of the groups were not entirely similar (all $p < 0.001$). Groups 1 and 3 showed no significant differences in age, N stage, M stage, or chemotherapy ($p = 0.467, 0.029, 0.277$, and $0.643$, respectively). Groups 2 and 4 exhibited no significant differences in age, GS, N stage, M stage, or RT ($p = 0.026, 0.250, 0.106, 0.277$, and $0.017$, respectively). However, Group 2 showed significantly higher T stage than the other groups (all $p < 0.001$).

**Table 1. Demographic and clinical characteristics of patients stratified by PSA level and GS.**

| Variable | Group 1 | Group 2 | Group 3 | Group 4 | Total | P-value |
|---|---|---|---|---|---|---|
| **N(%)** | 13304(3.6%) | 1741(0.5%) | 284732(77.9%) | 65781(18.0%) | 365558(100.0%) | |
| **Age** | | | | | | <0.001[a] |
| Median age, year (IQR) | 65.0 (59.0, 71.0) | 70.0 (64.0, 76.0) | 65.0 (59.0, 70.0) | 69.0 (63.0, 75.0) | 66.0 (60.0, 71.0) | |
| **Race, N(%)** | | | | | | <0.001[b] |
| Non-Black | 11986 (90.1%) | 1589 (91.3%) | 239057 (84.0%) | 56353 (85.7%) | 308985 (84.5%) | |
| Black | 1318 (9.9%) | 152 (8.7%) | 45675 (16.0%) | 9428 (14.3%) | 56573 (15.5%) | |
| **PSA** | | | | | | <0.001[b] |
| Median, ng/ml (IQR) | 1.6 (1.0, 2.1) | 1.7 (1.1, 2.2) | 6.5 (5.0, 9.4) | 10.6 (6.7, 21.2) | 6.7 (4.9, 10.4) | |
| **GS, N(%)** | | | | | | <0.001[c] |
| ≤6 | 8195 (61.6%) | 0 (0.0%) | 113788 (40.0%) | 0 (0.0%) | 121983 (33.4%) | |
| 7 | 5109 (38.4%) | 0 (0.0%) | 170944 (60.0%) | 0 (0.0%) | 176053 (48.2%) | |
| 8 | 0 (0.0%) | 751 (43.1%) | 0 (0.0%) | 31738 (48.2%) | 32489 (8.9%) | |
| 9 | 0 (0.0%) | 857 (49.2%) | 0 (0.0%) | 31062 (47.2%) | 31919 (8.7%) | |
| 10 | 0 (0.0%) | 133 (7.6%) | 0 (0.0%) | 2981 (4.5%) | 3114 (0.9%) | |
| **T stage, N(%)** | | | | | | <0.001 |
| 1 | 4884 (36.7%) | 423 (24.3%) | 130636 (45.9%) | 23086 (35.1%) | 159029 (43.5%) | |
| 2 | 7692 (57.8%) | 714 (41.0%) | 121831 (42.8%) | 23386 (35.6%) | 153623 (42.0%) | |
| 3 | 720 (5.4%) | 511 (29.4%) | 31625 (11.1%) | 17333 (26.3%) | 50189 (13.7%) | |
| 4 | 8 (0.1%) | 93 (5.3%) | 640 (0.2%) | 1976 (3.0%) | 2717 (0.7%) | |
| **N stage, N(%)** | | | | | | <0.001[a] |
| 0 | 13254 (99.6%) | 1557 (89.4%) | 280896 (98.7%) | 57680 (87.7%) | 353387 (96.7%) | |
| 1 | 50 (0.4%) | 184 (10.6%) | 3836 (1.3%) | 8101 (12.3%) | 12171 (3.3%) | |
| **M stage, N(%)** | | | | | | <0.001[a] |
| 0 | 13272 (99.8%) | 1572 (90.3%) | 283185 (99.5%) | 58849 (89.5%) | 356878 (97.6%) | |
| 1 | 32 (0.2%) | 169 (9.7%) | 1547 (0.5%) | 6932 (10.5%) | 8680 (2.4%) | |
| **Surgery** | | | | | | <0.001 |
| No | 5883 (44.2%) | 870 (50.0%) | 164269 (57.7%) | 43025 (65.4%) | 214047 (58.6%) | |
| Yes | 7421 (55.8%) | 871 (50.0%) | 120463 (42.3%) | 22756 (34.6%) | 151511 (41.4%) | |
| **Radiation therapy** | | | | | | <0.001[c] |
| No | 10603 (79.7%) | 872 (50.1%) | 192268 (67.5%) | 31048 (47.2%) | 234791 (64.2%) | |
| Yes | 2701 (20.3%) | 869 (49.9%) | 92464 (32.5%) | 34733 (52.8%) | 130767 (35.8%) | |
| **Chemotherapy** | | | | | | <0.001[d] |
| No/unknown | 13278 (99.8%) | 1667 (95.7%) | 284225 (99.8%) | 64000 (97.3%) | 363170 (99.3%) | |
| Yes | 26 (0.2%) | 74 (4.3%) | 507 (0.2%) | 1781 (2.7%) | 2388 (0.7%) | |

[a]There were no statistically significant differences observed between Group 1 and Group 3, as well as between Group 2 and Group 4.

[b]There were no statistically significant differences observed between Group 1 and Group 2.

[c]There were no statistically significant differences observed between Group 2 and Group 4.

[d]There were no statistically significant differences observed between Group 1 and Group 3.

Abbreviations: PSA = prostate-specific antigen; GS = Gleason score; N = number; IQR = interquartile range.

## Competing-risks regression analysis

In the univariate analysis, neither race nor RT significantly influenced PCSM. However, the multivariate analysis revealed that age markedly affected PCSM. Specifically, black individuals displayed higher HR for PCSM than did non-black individuals. Furthermore, Group 2 demonstrated a significantly increased risk of PCSM compared to the other groups. Regarding

the TNM classification, the HR for T1 stage was significantly lower than that for T3 and T4 stages but was higher than that for T2 stage. Similarly, the HR for N0 stage was significantly lower than that for N1 stage. Additionally, the HR for M0 stage was markedly lower than that for M1 stage. Both surgery and RT significantly reduced the risk of PCSM. The detailed results are presented in Table 2. In the univariate analysis of age, PSA and GS group, T stage, N stage, M stage, surgery, and chemotherapy, the Wald chi-squared test yielded p-values of less than 0.0001. In contrast, the univariate analyses for race and RT reported p-values of 0.4134 and 0.1637, respectively. In the multivariate analysis, the Wald chi-squared test resulted in a p-value of less than 0.0001.

Additionally, the cumulative incidence comparison revealed significantly higher PCSM in Group 2 than that in the other groups (all p < 0.00001), as illustrated in Fig 2. The PCSM in Group 4 was significantly greater than that in Groups 1 and 3 (both p < 0.00001). However, there were no significant differences in PCSM between Groups 1 and 3 (p = 0.060).

**Table 2. Univariate and multivariate competing-risks regression analysis for PCSM.**

| Variable | Univariate analysis | | | Multivariate analysis | | |
|---|---|---|---|---|---|---|
| | HR | 95%CI | P-value | HR | 95%CI | P-value |
| Age | 1.075 | 1.073-1.078 | <0.001 | 1.040 | 1.037-1.042 | <0.001 |
| Race, N(%) | | | | | | |
| Non-Black | 1.000 | referent | | 1.000 | referent | |
| Black | 0.978 | 0.927-1.032 | 0.413 | 1.250 | 1.180-1.324 | <0.001 |
| PSA & GS Group | | | | | | |
| 1 | 0.057 | 0.046-0.070 | <0.001 | 0.114 | 0.091-0.141 | <0.001 |
| 2 | 1.000 | referent | | 1.000 | referent | |
| 3 | 0.071 | 0.063-0.081 | <0.001 | 0.132 | 0.114-0.152 | <0.001 |
| 4 | 0.722 | 0.636-0.819 | <0.001 | 0.666 | 0.578-0.767 | <0.001 |
| T stage, N(%) | | | | | | |
| 1 | 1.000 | referent | | 1.000 | referent | |
| 2 | 0.803 | 0.768-0.840 | <0.001 | 0.924 | 0.878-0.973 | 0.003 |
| 3 | 1.633 | 1.549-1.723 | <0.001 | 1.213 | 1.132-1.299 | <0.001 |
| 4 | 16.782 | 15.460-18.217 | <0.001 | 2.201 | 1.972-2.457 | <0.001 |
| N stage, N(%) | | | | | | |
| 0 | 1.000 | referent | | 1.000 | referent | |
| 1 | 8.696 | 8.266-9.148 | <0.001 | 1.653 | 1.542-1.772 | <0.001 |
| M stage, N(%) | | | | | | |
| 0 | 1.000 | referent | | 1.000 | referent | |
| 1 | 31.184 | 29.855-32.573 | <0.001 | 7.256 | 6.799-7.745 | <0.001 |
| Surgery | | | | | | |
| No | 1.000 | referent | | 1.000 | referent | |
| Yes | 0.497 | 0.476-0.519 | <0.001 | 0.730 | 0.687-0.777 | <0.001 |
| Radiation therapy | | | | | | |
| No | 1.000 | referent | | 1.000 | referent | |
| Yes | 1.029 | 0.989-1.070 | 0.164 | 0.797 | 0.758-0.837 | <0.001 |
| Chemotherapy | | | | | | |
| No/unknown | 1.000 | referent | | 1.000 | referent | |
| Yes | 14.089 | 12.957-15.320 | <0.001 | 1.437 | 1.296-1.594 | <0.001 |

Abbreviation: PCSM = prostate cancer-specific mortality; HR = hazard ratio; CI = confidence interval; N = number; PSA = prostate specific antigen; GS = Gleason score.

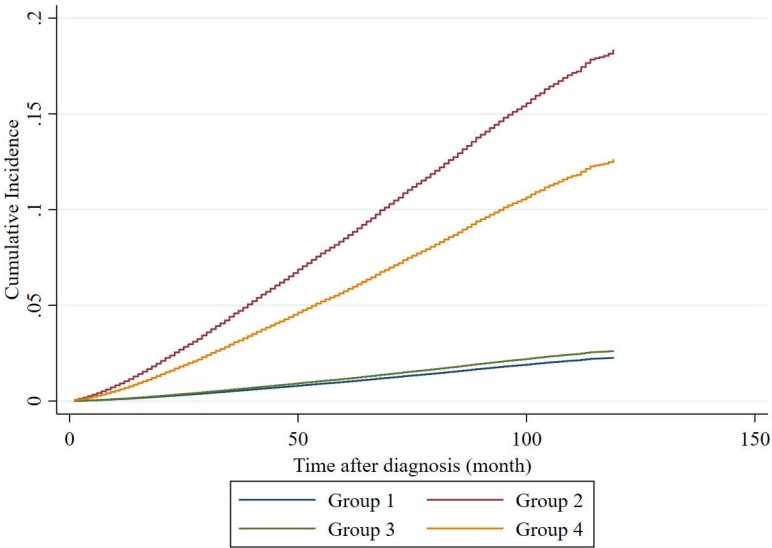

**Fig 2. Cumulative incidence function of PCSM across groups.** Abbreviation: PCSM = prostate cancer-specific mortality.

## Gene set enrichment analysis

In the GEO cohort, three samples were assigned to Group A and 95 to Group B, with their clinical data detailed in S1 Table. Initially, 75,148 Probe IDs were obtained. Out of these, 44,004 were successfully mapped to the corresponding gene symbols, which were then used for the GSEA. In Group B, no significantly upregulated gene sets were detected. In contrast, Group A demonstrated significant upregulation in 156 gene sets, as outlined in S2 Table. The three most significantly enriched gene sets were the translation initiation, insulin-like growth factor/insulin-like growth factor receptor/phosphoinositide 3-kinase (PI3K)/nuclear factor-κB (NF-κB), and epidermal growth factor (EGF)/epidermal growth factor receptor (EGFR)/PI3K/NF-κB signaling pathways. These gene sets exhibited the following normalized enrichment scores (NES) and significance values: NES = 2.623, p = 9.049E-08, adjusted p = 1.191E-05; NES = 2.491, p = 2.087E-06, adjusted p = 1.373E-04; and NES = 2.452, p = 3.187E-06, adjusted p = 1.398E-04 (Fig 3).

The clustering of significant pathways from the GSEA results revealed several enriched pathways (Fig 4), including polycomb repressive complex (PRC) 2, EGFR, erythroblastic oncogene B 2 (ERBB2), retrograde axonal transport, tumor necrosis factor (TNF)/NF-κB, Rho guanine nucleotide exchange factor (GEF)/RhoA and PI3K signaling pathways.

## Discussion

Our study found that PCa with low PSA levels and high grades was associated with higher tumor stage and an increased risk of PCSM compared to other PCa subtypes. In comparison with previous studies, our cohort included not only patients with cT1-4N0M0 PCa, but also those with regional lymph node metastasis and distant metastasis, making our analysis more comprehensive[6,7].

Another characteristic of this PCa subtype is its demonstrated resistance to ADT. Research by David Dewei Yang *et al.* indicated that in patients with localized or locally advanced PCa and GS ≥ 8, treated with definitive RT, ADT was associated with decreased ACM in patients

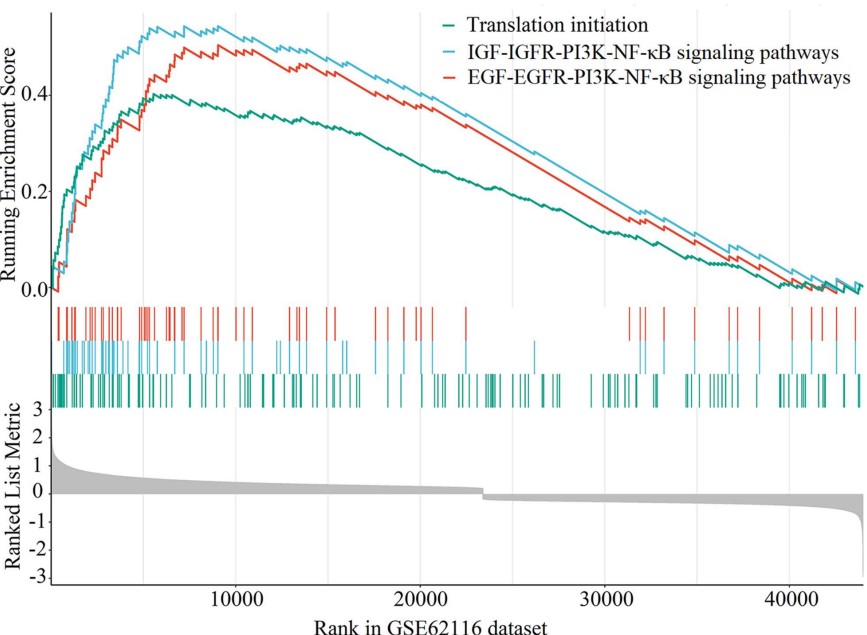

**Fig 3. Top three most significant gene sets identified in GSEA of the GEO Cohort.** Abbreviation: GSEA = Gene set enrichment analysis, GEO = Gene Expression Omnibus, IGF = insulin-like growth factor, IGFR = insulin-like growth factor receptor, PI3K = phosphoinositide 3-kinase, NF-κB = nuclear factor-κB, EGF = epidermal growth factor, EGFR = epidermal growth factor receptor.

with PSA > 2.5 ng/mL. In contrast, a trend toward increased ACM was observed in those with PSA ≤ 2.5 ng/mL. This result suggested that low-PSA, high-grade PCa may represent a distinct castration-resistant entity[13]. Similarly, research by Brandon A. Mahal *et al.* found that among patients with cT1-4N0M0 PCa and GS ≥ 8 who underwent RT, ADT conferred an overall survival benefit for those with PSA > 2.5 ng/mL, but not for those with PSA ≤ 2.5 ng/mL, further indicating resistance to ADT in this subgroup[6].

To improve the prognosis of patients with this PCa subtype, Brandon A. Mahal *et al.* suggested that the addition of docetaxel to the standard of care treatment for nonmetastatic PCa patients, who are in otherwise good health with PSA <4 ng/mL and GS ≥ 8, was associated with a significant reduction in PCSM[14]. Our study identified several significantly enriched pathways in this PCa subtype, and therapeutic agents targeting these pathways have been shown to exhibit divergent efficacy in inhibiting PCa progression.

The PI3K/ protein kinase B (AKT) signaling pathway plays a pivotal role in PCa progression by promoting cell survival through apoptosis inhibition, cell cycle acceleration, and enhanced proliferation[15]. Its activation is also associated with increased metastatic potential and resistance to ADT in PCa[16,17]. Inhibitors targeting this pathway have already shown efficacy in clinical trials. Capivasertib, a selective AKT inhibitor, has been demonstrated to enhance the anti-tumor effects of docetaxel by targeting residual docetaxel-resistant cells, irrespective of *PTEN* status, thereby inducing apoptosis and DNA damage[18]. In the phase II ProCAID clinical trial, the combination of Capivasertib and docetaxel improved survival in metastatic castration-resistant PCa (mCRPC) patients, regardless of *PTEN* status[19]. Additionally, another AKT inhibitor, ipatasertib, was evaluated in a phase III clinical trial, where the combination of ipatasertib and abiraterone significantly improved radiographic progression-free survival in mCRPC patients with *PTEN*-loss tumors compared to placebo

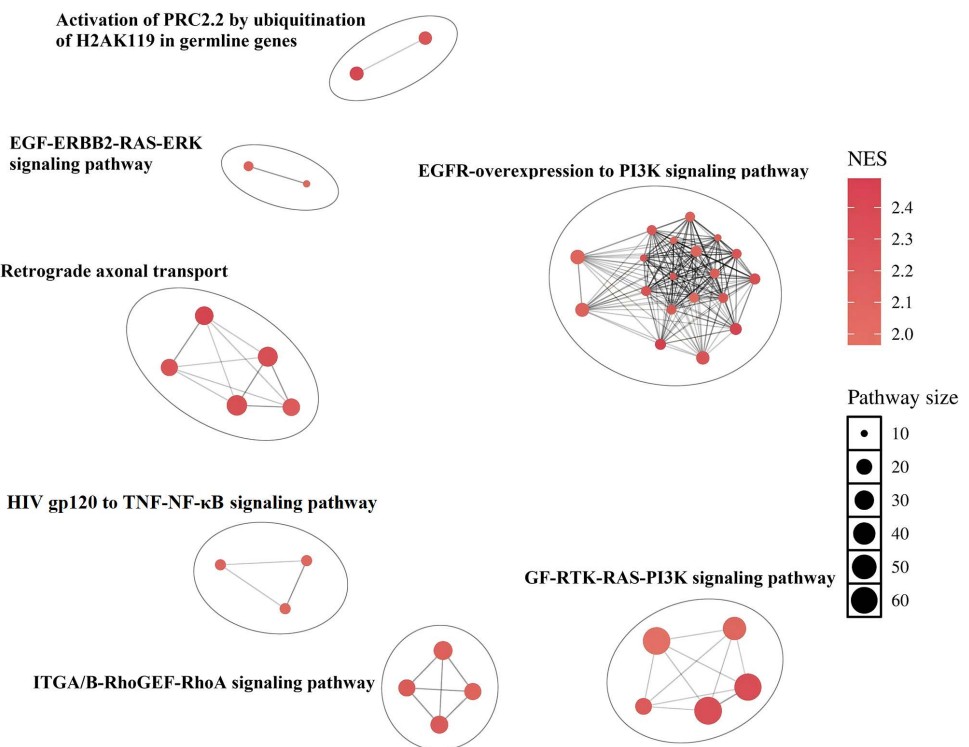

**Fig 4. Network visualization of enriched pathways based on GSEA results.** Abbreviation: GSEA = Gene set enrichment analysis, PRC = polycomb repressive complex, EGF = epidermal growth factor, ERBB2 = erythroblastic oncogene B 2, RAS = rat sarcoma virus, ERK = extracellular signal-regulated kinase, EGFR = epidermal growth factor receptor, PI3K = phosphoinositide 3-kinase, HIV = human immunodeficiency virus, TNF = tumor necrosis factor, NF-κB = nuclear factor-κB, GEF = guanine nucleotide exchange factor, GF = growth factor, RTK = receptor tyrosine kinase, NES = normalized enrichment scores.

plus abiraterone. However, no significant difference was observed in the intention-to-treat population[20].

Unlike PI3K/AKT inhibitors, statins act indirectly through metabolic pathways, offering a distinct therapeutic angle. Statins are competitive inhibitors of 3-hydroxy-3-methylglutaryl coenzyme A reductase, the rate-limiting enzyme in the mevalonate pathway responsible for cholesterol biosynthesis and its derivatives[21]. *In vitro* studies have demonstrated that statins promote apoptosis and inhibit PCa cell growth, primarily through the inactivation of RhoA signaling[22]. Moreover, when combined with caffeine, statins have been shown to downregulate phospho-AKT and anti-apoptotic proteins, thereby intersecting with the effects of PI3K/AKT inhibition[23]. Prospective and registry-based studies have also indicated a reduced risk of advanced and fatal PCa among statin users compared to non-users, as well as improved clinical outcomes in PCa patients receiving statin therapy[24]. Currently, a randomized, double-blind phase III clinical trial is underway to evaluate the efficacy of atorvastatin in patients with *de novo* metastatic PCa or high-risk M0 stage recurrent disease, for which ADT and antiandrogen therapy is initiated no longer than three months before recruitment[25].

While PI3K/AKT inhibitors act on signaling cascades, PRC2 inhibitors disrupt epigenetic reprogramming, offering complementary therapeutic avenues[26]. For instance, ORIC-944 has inhibited tumor growth in PCa xenograft models[27]. Additionally, it enhanced the antitumor activity of AR pathway inhibitors (ARPI) in preclinical combination studies and demonstrated *in vitro* synergy[27]. Currently, a phase I/Ib study is evaluating the efficacy of ORIC-944 as a

single agent or in combination with an ARPI for metastatic PCa[28]. Another PRC2 inhibitor, DZNeP, has demonstrated effectiveness against PCa cells *in vitro* and *in vivo*[29].

The EGFR/ERBB2 axis represents another AR-independent pathway, activating extracellular signal-regulated kinase and PI3K to promote PCa cell proliferation and metastasis[30,31]. In contrast to the above pathways, inhibitors targeting EGFR and ERBB2 have yielded limited clinical success. A multi-institutional, randomized phase II study evaluated gefitinib, an EGFR tyrosine kinase inhibitor, in patients with minimally symptomatic castration-resistant PCa. However, the treatment did not lead to significant reductions in PSA levels or measurable tumor responses[32]. Similarly, in another phase II multicenter clinical trial, lapatinib, a dual tyrosine kinase inhibitor targeting EGFR and ERBB2, was investigated in patients with castration-sensitive recurrent or metastatic PCa. This treatment also failed to demonstrate significant antitumor activity[33].

The NF-κB pathway plays a significant role in chemoresistance, radioresistance, and metastatic potential in PCa through cytokine-driven activation[34]. Thymoquinone and evodiamine attenuated tumor progression and migration in DU-145 models by suppressing NF-κB signaling[35,36]. Similarly, apigenin and bakuchiol reduced tumor proliferation and migration in PC-3 cells by blocking the NF-κB signaling[37,38]. While these compounds show preclinical efficacy, their clinical applicability remains to be validated.

Although previous studies have established a close relationship between nerve and the progression and metastasis of PCa, the specific molecules transported within the retrograde axonal transport signaling pathway remained unidentified[39]. As a result, further discussion on this aspect is currently not feasible.

In summary, inhibitors targeting the PI3K/AKT and RhoA signaling pathways have demonstrated substantial clinical potential in PCa treatment, supported by robust evidence from clinical trials. In contrast, therapeutic agents directed at PRC2 and NF-κB pathways remain in earlier developmental phases, with limited clinical validation to date. However, the efficacy of these inhibitors in improving outcomes for PCa patients with low PSA levels and high grades remains underexplored. Further rigorous investigation is warranted to elucidate their therapeutic utility in this distinct patient population.

Moreover, due to the critical role of PSA in early screening, patients with low-PSA, high-grade PCa may not be identified promptly[2,5]. Although these patients are at an increased risk of disease progression, they may not demonstrate biochemical recurrence[40]. Consequently, PSA may not be a reliable marker for detecting the progression of this disease either. It is essential to employ a comprehensive approach for early screening and monitoring of this PCa subgroup. This approach should incorporate multiple diagnostic methods, including digital rectal examination, clinical symptoms, alternative biomarkers, and advanced imaging techniques.

## Limitations

Our study has several limitations typical of population-based analyses. Chief among these are potential biases arising from incomplete data in the SEER database. Specifically, the dataset lacked information on medical comorbidities, cancer risk factors, performance status, chemotherapy regimens, and ADT. This absence constrained our ability to fully assess the impact of these variables on the study outcomes.

In addition, the follow-up period in our clinical cohort was relatively short. A longer-term follow-up period would likely provide deeper insights into patient prognosis and disease progression. Nevertheless, the aggressive nature of low-PSA, high-grade disease enabled the detection of significant differences within these limited follow-up intervals.

Another limitation is the GSE62116 dataset's lack of information on patient ethnicity, which could potentially introduce bias into our analysis. Moreover, the absence of patients with PSA ≤ 2.5 ng/mL and GS < 8 in the GSE62116 dataset prevented the comparison of gene expression profiles between patients with low PSA and high-grade disease and those with low PSA and low-grade disease. However, as the latter subgroup typically exhibited a better prognosis, our study focused on comparing patients with low PSA and high-grade disease and those with high PSA and high-grade disease.

## Conclusion

Patients with low-PSA, high-grade PCa typically exhibited significantly higher tumor stage, elevated PCSM and pronounced activation of multiple signaling pathways compared to other PCa subtypes. Due to the potential resistance to ADT in this subgroup, it is crucial to explore whether therapeutic agents targeting these pathways could improve the prognosis of patients with this specific PCa subtype. Furthermore, given their lower PSA levels, both early screening and post-treatment monitoring should emphasize physical examinations, other biomarkers, and imaging findings.

## Supporting information

**S1 Table. Clinical data of GEO cohort.**
(XLSX)

**S2 Table. Gene set enrichment analysis of the GEO cohort.**
(XLSX)

## Author contributions

**Conceptualization:** Ming Yang.

**Data curation:** Peng Zhang.

**Formal analysis:** Peng Zhang, Tieding Chen.

**Investigation:** Peng Zhang, Tieding Chen.

**Software:** Peng Zhang.

**Supervision:** Ming Yang.

**Writing – original draft:** Peng Zhang.

**Writing – review & editing:** Peng Zhang, Tieding Chen, Ming Yang.

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
