## [Decision Letter · Decision Letter 0]

17 Dec 2024

PONE-D-24-38111Clinical feature and gene expression analysis in low prostate-specific antigen, high-grade prostate cancerPLOS ONE

Dear Dr. Zhang,

Thank you for submitting your manuscript to PLOS ONE. After careful consideration, we feel that it has merit but does not fully meet PLOS ONE’s publication criteria as it currently stands. Therefore, we invite you to submit a revised version of the manuscript that addresses the points raised during the review process.

We look forward to receiving your revised manuscript.

Kind regards,

Maria Jesus Alvarez-Cubero

Academic Editor

PLOS ONE

Journal Requirements:

Additional Editor Comments :

Please take into account the authors comments to this article.

Reviewers' comments:

Reviewer's Responses to Questions

**Comments to the Author**

1. Is the manuscript technically sound, and do the data support the conclusions?

Reviewer #1: Yes

Reviewer #2: Yes

2. Has the statistical analysis been performed appropriately and rigorously? 

Reviewer #1: Yes

Reviewer #2: Yes

3. Have the authors made all data underlying the findings in their manuscript fully available?

Reviewer #1: No

Reviewer #2: Yes

4. Is the manuscript presented in an intelligible fashion and written in standard English?

Reviewer #1: Yes

Reviewer #2: Yes

5. Review Comments to the Author

Reviewer #1: This study addresses a critical area in prostate cancer (PCa) research, specifically the clinical characteristics and gene expression patterns in patients presenting with low prostate-specific antigen (PSA) levels but high-grade disease. This subgroup is particularly concerning, as these patients often exhibit resistance to standard treatments and poor outcomes.

The study's large sample size enhances the reliability of its findings, and the comprehensive statistical methods employed (Kruskal-Wallis H, Pearson's chi-squared tests, and competing-risks regression) bolster the validity of the results. The focus on gene expression adds an important molecular dimension to the clinical observations, offering potential avenues for targeted therapeutic strategies. However, the author(s) should address the following points:

1. It's important to specify the type of this cohort study in the methods section if it is retrospective or prospective.

2. While the findings are significant, the study would benefit from additional exploration of the mechanisms driving the observed gene expression changes.

3. This study requires more explanation of the PI3K pathway, focusing on its role in prostate cancer.

4. The full terms of PI3K, AKT/mTOR, IGF-IGFR-PI3K-NFKB, and EGF-EGFR-PI3K-NFKB should be write as they first mentioned in the manuscript.

Overall, this research contributes valuable insights into the clinical features and molecular underpinnings of low PSA, high-grade prostate cancer. The identification of the PI3K signaling pathway as a potential therapeutic target warrants further investigation and may ultimately lead to improved management strategies for this challenging patient population.

Further research could elucidate how these alterations contribute to treatment resistance and could guide the development of targeted therapies.

Reviewer #2: This paper looked into the clinical characteristics and gene expression mechanisms in prostate cancer patients with low PSA levels.

Accessing the SEER database, the authors categorized data concerning 365,558 patients into four groups, according to PSA levels and Grade Group. The paper reports that, compared to other subtypes, patients with low PSA levels and high-grade disease exhibited higher tumor stage and elevated prostate cancer-specific mortality, possibly via the activation of PI3K signaling pathway. The authors conclude that screening and post-treatment monitoring should include physical examination, imaging findings, and other biomarkers.

This study emphasized the urgent need for novel biomarkers and modalities with high sensitivity and specificity to facilitate prognostication and the development of individualized treatment options and follow-up strategies for prostate cancer patients. The research questions explored here – if repeated in a number of robust studies – can be of significance for the treatment and management of prostate cancer.

With that in mind, this reviewer wish the authors the very best with their research!

6. PLOS authors have the option to publish the peer review history of their article (what does this mean? ). If published, this will include your full peer review and any attached files.

**Do you want your identity to be public for this peer review?** For information about this choice, including consent withdrawal, please see our Privacy Policy .

Reviewer #1: No

Reviewer #2: No

---

## [Author Response · Author response to Decision Letter 0]

22 Dec 2024

Reviewer #1: This study addresses a critical area in prostate cancer (PCa) research, specifically the clinical characteristics and gene expression patterns in patients presenting with low prostate-specific antigen (PSA) levels but high-grade disease. This subgroup is particularly concerning, as these patients often exhibit resistance to standard treatments and poor outcomes.

The study's large sample size enhances the reliability of its findings, and the comprehensive statistical methods employed (Kruskal-Wallis H, Pearson's chi-squared tests, and competing-risks regression) bolster the validity of the results. The focus on gene expression adds an important molecular dimension to the clinical observations, offering potential avenues for targeted therapeutic strategies. However, the author(s) should address the following points:

1. It's important to specify the type of this cohort study in the methods section if it is retrospective or prospective.

2. While the findings are significant, the study would benefit from additional exploration of the mechanisms driving the observed gene expression changes.

3. This study requires more explanation of the PI3K pathway, focusing on its role in prostate cancer.

4. The full terms of PI3K, AKT/mTOR, IGF-IGFR-PI3K-NFKB, and EGF-EGFR-PI3K-NFKB should be write as they first mentioned in the manuscript.

Overall, this research contributes valuable insights into the clinical features and molecular underpinnings of low PSA, high-grade prostate cancer. The identification of the PI3K signaling pathway as a potential therapeutic target warrants further investigation and may ultimately lead to improved management strategies for this challenging patient population.

Further research could elucidate how these alterations contribute to treatment resistance and could guide the development of targeted therapies.

Response to Reviewer #1

We sincerely appreciate your valuable feedback, which helped us identify several shortcomings in the paper. In response, we have made the necessary revisions and added relevant content accordingly.

1. We emphasized the use of the SEER database and the GEO database for conducting a retrospective study on Page 6 Line 74, and Page 8 Line 116, respectively.

2. We analyzed the similarities between gene sets and summarized several enriched pathways, including the polycomb repressive complex 2, epidermal growth factor receptor, retrograde axonal transport, tumor necrosis factor/nuclear factor-κB, Rho guanine nucleotide exchange factor/RhoA, and phosphoinositide 3-kinase signaling (PI3K) pathways. We then discussed the impact of these pathways on prostate cancer (from Page 19 Line 251 to Page 22 Line 298) and outlined potential therapeutic strategies targeting them (from Page 22 Line 312 to Page 23 Line 322).

3. We have added a discussion on the role of the PI3K pathway in prostate cancer, highlighting its promotion of prostate cancer cell survival by inhibiting apoptosis, advancing cell cycle progression, and increasing proliferation rates. Additionally, we highlighted PI3K/protein kinase B signaling drives epithelial-mesenchymal transition, contributing to metastasis (from Page 16 Line 206 to Page 17 Line 211).

4. Thank you for your helpful suggestion. We have included the full names of the abbreviations for each pathway upon their first mention in the manuscript.

Reviewer #2: This paper looked into the clinical characteristics and gene expression mechanisms in prostate cancer patients with low PSA levels.

Accessing the SEER database, the authors categorized data concerning 365,558 patients into four groups, according to PSA levels and Grade Group. The paper reports that, compared to other subtypes, patients with low PSA levels and high-grade disease exhibited higher tumor stage and elevated prostate cancer-specific mortality, possibly via the activation of PI3K signaling pathway. The authors conclude that screening and post-treatment monitoring should include physical examination, imaging findings, and other biomarkers.

This study emphasized the urgent need for novel biomarkers and modalities with high sensitivity and specificity to facilitate prognostication and the development of individualized treatment options and follow-up strategies for prostate cancer patients. The research questions explored here – if repeated in a number of robust studies – can be of significance for the treatment and management of prostate cancer.

With that in mind, this reviewer wish the authors the very best with their research!

Response to Reviewer #2

During the revision process, we also identified several pathways, in addition to the PI3K pathway, which may be associated with poor prognosis of prostate cancer cases with low PSA levels and high-grade tumors. We hope these findings will contribute to improving the prognosis for these patients.

---

## [Editor Report · Decision Letter 1]

7 Jan 2025

PONE-D-24-38111R1Clinical feature and gene expression analysis in low prostate-specific antigen, high-grade prostate cancerPLOS ONE

Dear Dr. Zhang,

Thank you for submitting your manuscript to PLOS ONE. After careful consideration, we feel that it has merit but does not fully meet PLOS ONE’s publication criteria as it currently stands. Therefore, we invite you to submit a revised version of the manuscript that addresses the points raised during the review process.

You should improve discussion section including compare and contrast ideas and reinforcing why your results are so important.Change genes names into itatics as well as latin words.

We look forward to receiving your revised manuscript.

Kind regards,

Maria Jesus Alvarez-Cubero

Academic Editor

PLOS ONE

Additional Editor Comments:

Dear authors,

You should improve discussion section, comparing and contrast the information, results that you suggest. In my opinion it is strange to see "hypothesis senteces" in this section. Moreover, you should pay attention to genes names and latin words that should be in italics.

---

## [Author Response · Author response to Decision Letter 1]

4 Feb 2025

We greatly appreciate the reviewer’s insightful feedback, and we have revised the manuscript accordingly.

1. Significance of the Study: First, our study identifies that prostate cancer (PCa) patients with low PSA levels and high-grade tumors tend to present with higher tumor stages and a significantly increased risk of prostate cancer-specific mortality (PCSM). This underscores the need for heightened clinical attention for these patients. Second, prior studies have also highlighted that this PCa subtype exhibits resistance to androgen deprivation therapy (ADT). Therefore, improving the prognosis of these patients is of paramount importance. Our findings contribute valuable insights toward addressing this issue. Specifically, we identified several enriched signaling pathways in this patient population. Existing research has shown that targeting these pathways with therapeutic agents can yield varying degrees of efficacy in treating PCa. Consequently, future studies should investigate the potential of these targeted therapies in improving outcomes for patients with low-PSA, high-grade PCa. Both of these points reinforce the significance of our research.

Modifications to the Discussion: We have made appropriate revisions to the discussion section, condensing some content, expanding on others, and re-organizing the structure for clarity. First, compared to previous studies, our research included patients with regional lymph node metastasis and distant metastasis, providing a more comprehensive cohort. Second, we emphasize that patients with low PSA levels and high-grade PCa exhibit resistance to ADT, highlighting the need for alternative treatment strategies. Lastly, we discuss the enriched signaling pathways identified in our study and the potential therapeutic agents, suggesting that further research is required to validate their efficacy in low-PSA, high-grade PCa.

2. We have carefully revised the manuscript to address the reviewer’s suggestion. All gene names, including PTEN, and Latin terms have been italicized in accordance with the requested formatting guidelines.

---

## [Editor Report · Decision Letter 2]

13 Feb 2025

PONE-D-24-38111R2Clinical feature and gene expression analysis in low prostate-specific antigen, high-grade prostate cancerPLOS ONE

Dear Dr. Zhang,

Thank you for submitting your manuscript to PLOS ONE. After careful consideration, we feel that it has merit but does not fully meet PLOS ONE’s publication criteria as it currently stands. Therefore, we invite you to submit a revised version of the manuscript that addresses the points raised during the review process.

**In my opinion, the manuscript still needs changes such as, indicating why you use GG classification? This is not an extended way to name clinical classification in PCa. Why do not change for universal classification?****According to discussion, it is true that is improved, but including subheading in this part is not common and make the text less fluid, so I will suggest to connect comparing and contrasting all these sections.**

We look forward to receiving your revised manuscript.

Kind regards,

Maria Jesus Alvarez-Cubero

Academic Editor

PLOS ONE

**Additional Editor Comments:**

In my opinion, the manuscript still needs changes such as, indicating why you use GG classification? This is not an extended way to name clinical classification in PCa. Why do not change for universal classification?

According to discussion, it is true that is improved, but including subheading in this part is not common and make the text less fluid, so I will suggest to connect comparing and contrasting all these sections.

Best regards,

---

## [Author Response · Author response to Decision Letter 2]

22 Feb 2025

We greatly appreciate the reviewer’s insightful feedback, and we have revised the manuscript accordingly.

1. We thank the reviewers for their astute observation regarding grading terminology. In response to your suggestion, we have replaced "Grade Group" with "Gleason score" throughout the manuscript. Additionally, we have made corresponding revisions to the relevant content and updated the data in the tables to reflect this change.

2. We are grateful for the reviewers` insightful critique of our pathway-focused approach. In our study, we identified several enriched signaling pathways in prostate cancer (PCa) patients with low PSA levels and high grades. Given that these patients may resist androgen deprivation therapy, we hypothesized that pharmacological agents targeting these pathways could potentially improve clinical outcomes. To substantiate this hypothesis, we systematically compiled and compared evidence-based data from clinical trials for drugs associated with each pathway.

Our analysis revealed that inhibitors targeting the PI3K/AKT and RhoA signaling pathways have shown substantial clinical promise in PCa treatment, supported by robust trial data. Conversely, therapeutic agents directed at the PRC2 and NF-κB pathways remain in earlier developmental stages, with limited clinical validation to date. These findings highlight prioritized directions for future clinical investigations.

During the revision process, we streamlined the functional descriptions of these pathways to enhance the manuscript’s readability and focus, ensuring that the core findings remain central to the narrative.

---

## [Editor Report · Decision Letter 3]

11 Mar 2025

Clinical feature and gene expression analysis in low prostate-specific antigen, high-grade prostate cancer

PONE-D-24-38111R3

Dear Dr.Peng Zhang,

We’re pleased to inform you that your manuscript has been judged scientifically suitable for publication and will be formally accepted for publication once it meets all outstanding technical requirements.

Kind regards,

Maria Jesus Alvarez-Cubero

Academic Editor

PLOS ONE
---

## [Editor Report · Acceptance letter]

PONE-D-24-38111R3

PLOS ONE

Dear Dr. Zhang,

I'm pleased to inform you that your manuscript has been deemed suitable for publication in PLOS ONE. Congratulations! Your manuscript is now being handed over to our production team.

Kind regards,

on behalf of

Dr. Maria Jesus Alvarez-Cubero

Academic Editor

PLOS ONE